# Extraction, Structural Analysis, and Biofunctional Properties of Exopolysaccharide from *Lactiplantibacillus* *pentosus* B8 Isolated from Sichuan Pickle

**DOI:** 10.3390/foods11152327

**Published:** 2022-08-04

**Authors:** Guangyang Jiang, Ran Li, Juan He, Li Yang, Jia Chen, Zhe Xu, Bijun Zheng, Yichen Yang, Zhongmei Xia, Yongqiang Tian

**Affiliations:** 1College of Biomass Science and Engineering, Sichuan University, Chengdu 610065, China; 2Key Laboratory of Leather Chemistry and Engineering, Ministry of Education, Sichuan University, Chengdu 610065, China; 3College of Food Science, Sichuan Agricultural University, Ya’an 625014, China; 4Key Laboratory of Bio-Resources and Eco-Environment of Ministry of Education, College of Life Sciences, Sichuan University, Chengdu 610065, China; 5Institute of Biotechnology and Nucleic Technology, Sichuan Academy of Agricultural Sciences, Chengdu 610066, China

**Keywords:** exopolysaccharide, structural analysis, biofunctional properties, *Lactiplantibacillus pentosus*

## Abstract

Two novel exopolysaccharides, named LPB8-0 and LPB8-1, were isolated and purified from *Lactiplantibacillus pentosus* B8. Moreover, their structure and bioactivities were evaluated through chemical and spectral means. The study results demonstrated that LPB8-0 was primarily composed of mannose and glucose and had an average molecular weight of 1.12 × 10^4^ Da, while LPB8-1 was composed of mannose, glucose, and galactose and had an average molecular weight of 1.78 × 10^5^ Da. Their carbohydrate contents were 96.2% ± 1.0% and 99.1% ± 0.5%, respectively. The backbone of LPB8-1 was composed of (1→2)-linked α-D-Man*p* and (1→6)-linked α-D-Man*p*. LPB8-0 and LPB8-1 had semicrystalline structures with good thermal stability (308.3 and 311.7 °C, respectively). SEM results displayed that both LPB8-0 and LPB8-1 had irregular thin-slice shapes and spherical body structures. Additionally, an emulsifying ability assay confirmed that LPB8-0 and LPB8-1 had good emulsifying activity against several edible oils, and this activity was retained under acidic, neutral, and high temperature conditions. Furthermore, an antioxidant assay confirmed that LPB8-1 had stronger scavenging activity than LPB8-0. Overall, these results provide a theoretical basis for the potential application of these two novel exopolysaccharides as natural antioxidants and emulsifiers in the food and pharmaceutical industries.

## 1. Introduction

Exopolysaccharides (EPSs), produced by various microorganisms (bacteria, fungi, and microalgae) during their growth phases, are high-molecular-weight and structurally diverse groups of natural biomacromolecules [1]. EPSs have recently attracted considerable attention from many researchers because of their potential applications in various industries. Compared with artificial polymers, natural EPSs are almost inexhaustible polymers, as they are not dependent on external environmental conditions.

Lactic acid bacteria (LAB) have been generally recognized as safe (GRAS) by the FDA, and LAB-EPSs have also been recognized as safe agents. Previous studies have reported that LAB-EPSs from different sources possess multiple functional bioactive activities, such as antioxidation, anticancer, immunomodulatory, and prebiotic activities [2,3,4]. Unlike other polymers, LAB-EPSs have attracted increasing attention because of their unique physicochemical properties and comprehensive applications in some industrial fields, including food additives, pharmaceuticals, and wastewater treatment, as an emulsifying agent, coagulant, thickener, flocculant, etc. Indeed, the bioactive activities and physicochemical properties of EPSs have been verified to be closely associated with their chemical structures and complexity, including their monosaccharide composition, molecular weight (MW), linkage type, and substituent groups [5]. Consequently, a systemic characterization of the structure of various LAB-EPSs is of high significance to exploiting their functional properties and providing a theoretical basis for their future applications in various industries.

So far, numerous LAB-EPS-producing strains have been screened and identified from traditional fermented foods. For instance, LAB-EPSs from *Lactobacillus rhamnosus* ZFM231 [6], *Lactobacillus plantarum* CNPC003 [7], and *Streptococcus thermophilus* ZJUIDS-2-0 [8] have been reported to possess potential bioactivities and functional properties. As a representative of fermented vegetables, Sichuan pickle is a LAB-rich resource. Notably, *Lactiplantibacillus* spp. (gram-positive, facultatively anaerobic, non-motile, and non-spore-forming) plays an indispensable role in pickle fermentation. Additionally, the EPS from *Lactiplantibacillus plantarum* MM89 exhibited excellent immunomodulatory activity and hence could be regarded as a convenient additive or functional immunomodulatory agent for food products [3]. A new EPS-producing strain, *L. pentosus* B8, was recently screened from Sichuan pickle in our laboratory. So far, very few studies have verified the structural characteristics and biofunctional properties of the EPSs from *L. pentosus.*

The present work sought to isolate EPSs from *L. pentosus* B8. Furthermore, the structural characteristics, emulsifying activities, and antioxidant activities of these EPSs were assessed.

## 2. Materials and Methods

### 2.1. Bacterial Propagation

An EPS-producing strain was isolated from Sichuan pickle. Based on its morphological and physiological characteristics and 16S rDNA sequence analysis, the strain was identified as *L. pentosus* (GenBank accession number: MW898221) and named as strain B8. This strain was routinely grown on MRS broth at 30 °C and preserved in 30% (*v*/*v*) glycerol.

### 2.2. Crude EPS Extraction and Purification

The EPS of *L. pentosus* B8 was obtained according to a previously reported method, with minor modifications [9]. Briefly, *L. pentosus* B8 was inoculated in MRS medium supplemented with 40 g/L sucrose and incubated at 30.0 ± 0.1 °C for 48 h. Subsequently, the fermented broth was centrifuged (12,000× *g*, 10 min), and the supernatant was added to 95% (*v*/*v*) cold ethanol. The precipitates were resuspended in ultrapure water, and the protein was removed using an enzyme combined with the Savage method [10]. Briefly, papain (800 U/mL) was mixed with the precipitate solution (pH 6.0) and maintained at 55 °C (water bath) for 100 min, followed by thorough mixing with chloroform and n-butanol (4:1, *v*/*v*). Later, this mixture was centrifuged, dialyzed (MW: 8–14 kDa), and lyophilized to obtain the crude EPS. Using an anion-exchange chromatographic column (DEAE-52; GE Healthcare, Stockholm, Sweden), the sample was eluted with ultrapure water and three NaCl concentrations (0.1, 0.3, and 0.5 M NaCl). The fractions were collected from 4 mL aliquots per tube, and then the total sugar content was measured using the phenol–sulfuric acid method. The primary fractions were pooled, dialyzed, and then lyophilized. Afterward, the lyophilized EPS sample was redissolved in ultrapure water at a final concentration of 30 mg/mL. Further EPS purification was performed on Sephacryl S-300 HR and S-400 HR columns (1.6 cm × 90 cm; GE Healthcare, Stockholm, Sweden) and eluted with ultrapure water at a flow rate of 1 mL/min. Finally, the resulting fractions were pooled and freeze-dried for structural characterization and functional evaluation.

### 2.3. Chemical Composition Characterization Assays

The total sugar content of the EPSs was estimated by the phenol–sulfuric acid method by using glucose as a standard. The protein content was estimated using the bicinchoninic acid (BCA) method. The EPSs (1 mg/mL) were recorded in the range of 200–800 nm by using a UV–vis spectrophotometer (Hitachi, Tokyo, Japan) to evaluate the presence of protein or nucleic acid.

### 2.4. Determination of MW

The MWs of the EPSs were determined through high-performance size-exclusion chromatography (HPSEC) along with refractive index (RI) (Optilab T-Rex, Wyatt Technology, Santa Barbara, CA, USA) and multiangle laser light scattering (MALLS) detectors (DAWN HELEOS II, Wyatt Technology, Santa Barbara, CA, USA). The EPS samples were dissolved in 0.1 M NaNO_3_ aqueous solution (5 mg/mL) and filtered through a filter (0.45 μm). The LPB8-0 and LPB8-1 solutions (5 mg/mL, 100 mL) were added to the HPSEC–RI–MALLS system. The EPSs were eluted with 0.1 mol/L NaNO_3_ solution.

### 2.5. Monosaccharide Composition

The monosaccharides of LPB8-0 and LPB8-1 were determined through high performance anion-exchange chromatography with pulsed amperometric detection (HPAEC-PAD). The LPB8-0 and LPB8-1 samples (5.0 mg) were hydrolyzed with trifluoroacetic acid (2.0 M) at 121 °C for 2 h. Then, two hydrolysates were evaporated to dryness under an N_2_ stream blowing instrument and eluted with methanol. The released monomers and all standards were further measured using a Thermo ICS-5000 ion chromatography system (Thermo Scientific, Waltham, MA, USA) fitted with a Dionex CarboPac PA-20 analytical column and a Dionex ED50A electrochemical detector.

### 2.6. Linkage Analysis by Methylation 

The linkage analysis method of Nasir et al. [11] was used, with minor modifications. In total, 10 mg of the LPB8-1 sample was dissolved in anhydrous dimethyl sulfoxide and methylated with 0.5 mL methyl iodine. After methylation, the methylated sample was hydrolyzed with TFA, reduced with NaBD4, and acetylated to produce the derivative for a 7890A-5977B GC–MS system (Agilent Technologies, Palo Alto, CA, USA) equipped with an HP-5MS capillary column. The program was isothermal at 140 °C; the hold time was 2 min, with a temperature gradient of 3 °C/min up to a final temperature of 230 °C.

### 2.7. FTIR and NMR Spectroscopy Analysis

FTIR spectroscopy (Thermo Fisher Scientific, Waltham, MA, USA) was used to determine the major chemical groups of the EPSs. The IR spectra were recorded from 600 to 4000 cm^−1^. Then, a 40 mg/mL solution of the sample was prepared with 99.9% D2O (500 μL) as the solvent. The 1D and 2D NMR data were recorded using a Bruker spectrometer (600 MHZ, Bruker, Karlsruhe, Switzerland).

### 2.8. Thermal Stability Evaluation

The thermal properties of the EPSs (LPB8-0 and LPB8-1) were measured by a thermogravimetric analyzer (209F3, NETZSCH, Free State of Bavaria, Germany). Approximately 5 mg of each of the EPS samples were placed in a standard aluminum pan and heated between 50 and 800 °C in a nitrogen atmosphere (10 °C/min).

### 2.9. Examination of X-ray Diffractometry, Particle Size, and Zeta Potential 

The X-ray diffraction (XRD) data of LPB8-0 and LPB8-1 were analyzed using an X-ray diffractometer (D8 ADVANCE Bruker, Germany). The LPB8-0 and LPB8-1 samples were recorded at 2θ angles from 5 to 85° with a scanning rate of 10°/min. The size distributions and zeta potentials of the EPSs (0.5% *w*/*v*) were measured using a NanoPlus Zeta Potential and Particle Size Analyze*r* (ZEN5600, Malvern Instruments, Malvern, UK) at 25 °C. The LPB8-0 and LPB8-1 samples were dissolved in ultrapure water.

### 2.10. Scanning Electron Microscopy (SEM) and Atomic Force Microscopy (AFM)

The morphological characteristics of LPB8-0 and LPB8-1 samples were observed using a Apreo 2C SEM (Thermo Fisher Scientific, Waltham, MA, USA) at a voltage of 15.0 kV with magnification of 1000× and 5000×, respectively. Then, 10 μg/mL of EPS solution was deposited onto the surface of freshly mica plate and was allowed to air-dry. The tapping mode was used to acquire the topographies of LPB8-0 and LPB8-1 by using an atomic force microscope (TESPA-V2, Bruker, Billerica, MA, USA), and the AFM image size was selected as 8 μm × 8 μm.

### 2.11. Emulsifying Activities of the EPSs

The emulsifying activities of LPB8-0 and LPB8-1 were measured according to our previously reported methods [12]. Briefly, 3 mL of several edible oils (soybean oil, coconut oil, olive oil, corn oil, rap oil, peanut oil, palm oil, and sunflower oil) were added to 2 mL of the EPS solutions at a concentration of 1 mg/mL. Then, each sample was stirred for 5 min. Later, the effects of different EPS concentrations (0.1, 0.5, 1.0, 1.5, and 2.0 mg/mL), temperatures (25, 40, 60, 80, and 100 °C), and pH values (4.0, 6.0, 8.0, 10.0, and 12.0) on emulsion stability were assessed. The emulsifying activity (EA, %) was analyzed after 1, 24, and 168 h using the following equation:EA (%)=(emulsion layer height/total height)×100

### 2.12. Analysis of the Antioxidant Activities of LPB8-0 and LPB8-1 In Vitro

The antioxidant activities (DPPH radical scavenging ability, ABTS^+^ free radical scavenging ability, and hydroxyl free radical scavenging ability) of LPB8-0 and LPB8-1 were evaluated according to the previously reported method [13].

#### 2.12.1. DPPH Scavenging Activity

Briefly, approximately 100 μL of LPB8-0 and LPB8-1 sample solution of different concentrations were immersed in 10 mL DPPH solution. Then, the mixture was incubated for 30 min in darkness. The absorbance was recorded at 517 nm on a UV–visible spectrophotometer (PerkinElmer, Waltham, MA, USA). The DPPH radical scavenging ability of samples was calculated using the following formula:$$\mathrm{scavenging}\;\mathrm{ability}\;\left(\%\right)=\left(1-A_{sample}\right)/A_{blank}\times 100$$
where $A_{sample}$ and $A_{blank}$ represent the absorbances of the test and control samples, respectively. 

#### 2.12.2. ABTS Radical Scavenging Activity

For ABTS radical scavenging activity, equal volumes of ABTS solution (7 mmol/L) and potassium persulfate solution (2.45 mmol/L) were mixed and placed at room temperature overnight. Then, the stock solution was added to various concentrations of LPB8-0 and LPB8-1 samples. The reaction mixture was reacted at 37 °C for 30 min. The absorbance of the samples was measured at 732 nm, and the ABTS radical scavenging activity was determined as follows:$$\mathrm{scavenging}\;\mathrm{ability}\;\left(\%\right)=\left(1-A_{sample}\right)/A_{blank}\times 100$$
where $A_{sample}$ is the absorbance of a sample and $A_{blank}$ is the absorbance of a blank.

#### 2.12.3. Hydroxyl Free Radical Scavenging Activity

Then, 1.0 mL of LPB8-0 and LPB8-1 samples at different concentrations were blended with FeSO_4_ (2 mmol/L) and salicylic acid—ethanol solution (6 mmol/L), and then, H_2_O_2_ solution (0.5 mL, 9 mmol/L) was added to the mixed solutions before incubation at 37 °C for 30 min. Subsequently, the absorbance of the mixtures was immediately tested at 536 nm. The scavenging activity of LPB8-0 and LPB8-1 samples was calculated as follows:$$\mathrm{scavenging}\;\mathrm{ability}\;\left(\%\right)=1-\left(A_{2}-A_{1}\right)/A_{0}\times 100$$
where $A_{0}$, $A_{1}$, and $A_{2}$ represent the absorbance of the reagent blank, control, and samples, respectively.

### 2.13. Statistical Analysis

Statistical analysis and graph plotting were performed using the SPSS 20.0 (IBM Co., Armonk, NY, USA) and Origin 9.0.0 software (Origin Lab Co., Northampton, MA, USA), and values are expressed as means ± standard deviations.

## 3. Results and Discussion

### 3.1. Extraction, Purification, and Chemical Composition of the EPSs

The crude EPSs of *L. pentosus* B8 were harvested through a series of processing steps, including ethanol precipitation, deproteinization, dialysis, and lyophilization. The EPS production yield was 1401.52 ± 9.54 mg/L. It was first isolated using a DEAE-52 anion-exchange column, and two fractions (designated as LPB8-0 and LPB8-1, respectively) displayed on the elution profile were obtained (Figure 1A). Then, LPB8-0 and LPB8-1 were purified through Sephacryl S-300 HR and S-400 HR gel-filtration columns, respectively. As depicted in Figure 1B,C, the elution profiles of the two fractions appeared with only one single peak each, showing that LPB8-0 and LPB8-1 were homogeneous. Further dialysis and lyophilization produced LPB8-0 and LPB8-1 fractions with a purity of 96.2% ± 1.0% and 99.1% ± 0.5%, but sulfate was not detected in the samples (Table 1). Compared with the crude EPS, no obvious absorption appeared at 260 or 280 nm in the UV–vis spectra (Figure 2A), indicating no nucleic acids or proteins in LPB8-0 and LPB8-1.

### 3.2. MW Distribution and Monosaccharide Composition Analysis of the EPSs

The MWs of LPB8-0 and LPB8-1 were tested using an HPSEC–RI–MALLS system. As tabulated in Table 1, the MWs of LPB8-0 and LPB8-1 were 1.12 × 10^4^ and 1.78 × 10^5^ Da, respectively. This phenomenon indicated that one strain could produce EPSs of different MWs. These results were consistent with those of a previous study reporting LAB-EPSs in the MW range of 10^4^–10^6^ g/mol [14]. 

The monosaccharide composition of LPB8-0 and LPB8-1 was detected using the HPAEC-PAD system (Figure 2B). LPB8-0 was primarily composed of approximately 15.76% mannose and 84.24% glucose, while LPB8-1 was composed of mannose, glucose, and galactose at the molar ratios of 77.74%, 21.08%, and 1.18%, respectively. Notably, no uronic acid was found in LPB8-0 and LPB8-1. The yield, total carbohydrate content, and monosaccharide composition of LPB8-1 were higher than those of LPB8-0, which was used in the subsequent experiments.

### 3.3. Functional Groups and Glycosidic Linkages

As they represent the most important and reliable analytical method, FTIR absorption spectra were employed to analyze the major characteristic peaks and linkage bonds in the samples. As depicted in Figure 2C, the obvious peaks at 3382 (LPB8-1) and 3403 (LPB8-0) cm^−1^ represented the O–H stretching vibration. The peaks at 2932 (LPB8-1) and 2929 (LPB8-0) cm^−1^ were caused by the stretching vibration of C–H. The absorption peaks at 1645 (LPB8-1) and 1641 (LPB8-0) cm^−1^ might have been due to the associated water, and the peaks at 1412 cm^−1^ could be assigned to the bending vibration of C–OH in this sample [15]. Notably, the characteristic absorption bands in the range of 900−1200 cm^−1^ were attributed to the stretching vibrations of C−O−C and C−O−H [16]. Weak peaks were observed from 800 to 900 cm^−1^, indicating the presence of α- and β-configurations.

LPB8-1 was subjected to methylation treatment followed by GC–MS analysis to confirm the types of glycosidic linkages of monosaccharide residues. The results are summarized in Table 2 and Appendix A. The reduced LPB8-1 exhibited ten types of linkages, including T-Man*p*-(1→, →3)-Man*p*-(1→, →2)-Man*p*-(1→, →6)-Man*p*-(1→, →6)-Glc*p*-(1→, →4)-Glc*p*-(1→, →3,6)-Man*p*-(1→, →2,6)-Man*p*-(1→, →2,6)-Glc*p*-(1→, and →2,3,6)-Man*p*-(1→ with molar ratios of 36.00:11.94:17.31:2.58:2.93:11.11:0.97:10.10:6.74:0.32, respectively.

### 3.4. NMR Analysis

A deep structural characterization of LPB8-1 was investigated with 1D (1D-1H, 1D-13C) and 2D (H-^1^H COSY, TOCSY, and NOESY; ^1^H-^13^C HSQC and HMBC) NMR data to obtain more comprehensive information about the linkages between the glycosyl residues and signal assignments. As depicted in the ^1^H NMR spectrum (Figure 3A), a group of the predominant signals (H-1) ranging from δ 4.52 to 5.52 ppm was observed. Intensive anomeric carbon signals (C-1) were located at δ 99.30–104.69 ppm in the ^13^C NMR spectrum (Figure 3B). The broad signals between δ 3.30 and 4.33 ppm were related to H-2–H-6 signals, and the corresponding carbon signals at δ 61.42–80.99 ppm were the characteristic peaks of C-2–C-6. A total of eleven anomeric protons (5.43, 5.33, 5.19, 5.18, 5.16, 5.13, 5.09, 5.08, 5.05, 4.93, and 4.56 ppm) appeared in the ^1^H spectrum, which are labelled with A–K, respectively. The chemical shifts of anomeric proton and carbon signals were confirmed through the 2D NMR (COSY, HSQC, and HMBC) spectrum analysis.

Based on the eleven sugar signals, a representative signal of residue B was extensively investigated. Residue B had a relatively intensive anomeric proton signal at δ 5.33 ppm, and its corresponding anomeric carbon signal at δ 102.29 ppm was successfully obtained through HSQC spectrum analysis (Appendix A), showing the presence of an α-configuration. As shown in Figure 3C and Appendix A, further combining of the COSY and TOCSY spectral data revealed that other proton signals at δ 5.33/4.16 ppm, 4.16/3.96 ppm, 3.96/3.79 ppm, 3.79/3.92 ppm, and 3.92/3.80 (3.66) ppm were mainly caused by the H-1/H-2, H-2/H-3, H-3/H-4, H-4/H-5, and H-5/H-6, respectively, of residue E. Hence, the proton signals of residue B occurred at δ 4.16 ppm, 3.96 ppm, 3.79 ppm, 3.92 ppm, and 3.80 (3.66) ppm for H-2–H-6, respectively. According to the HSQC spectrum of LPB8-1, the strong cross-peaks of 5.33/102.29, 4.16/79.80, 3.96/71.68, 3.79/68.01, 3.92/74.49, and 3.80 (3.66)/62.38 were attributed to the H-1/C-1, H-2/C-2, H-3/C-3, H-4/C-4, H-5/C-5, and H-6/C-6, respectively. Compared with a previous study, the C-2 (δ 79.80 ppm) moved downward, showing that it was substituted at the position of C-2 [17,18]. Hence, residue B was identified as →2)-α-D-Man*p*-(1→. Subsequently, the signal changes of other residues were identified and deduced using the same method, and the detailed signal assignments of protons and carbon are summarized in Table 3 [19,20,21,22].

Afterward, the correlations of the linkage sites/types and sequences between different sugar residues in LPB8-1 were evaluated, and the NOESY and HMBC spectra were analyzed (Appendix A). In the HMBC spectrum, the cross-peak between the H-1 of residue B (δ 5.33 ppm) and the C-5 of residue F (δ 74.57 ppm) indicated that H-1 was related to O-6 and O-2, suggesting the existence of →2)-α-D-Man*p*-(1→2,6)-α-D-Man*p*-(1→. Similarly, the H-1 of residue D (δ 5.18 ppm) was found to be linked to the O-6 of residue F (δ 74.57 ppm), suggesting the presence of α-D-Man*p*-(1→2,6)-α-D-Man*p*-(1→. The H-1 signals of α-D-Man*p*-(1→ (G) at δ 5.09 ppm corresponded to the C-2 signal of →2)-α-D-Man*p*-(1→(B) at δ 79.80 ppm, demonstrating the linkage α-D-Man*p*-(1→2)-α-D-Man*p*-(1→. In the NOESY spectrum, the correlations at 5.09/4.16, 5.08/4.16, and 4.56/3.78 were assigned to G (H-1)/H (H-2), H (H-1)/B (H-2), and K (H-1)/E (H-6), respectively. These related signals confirmed the existence of α-D-Man*p*-(1→2)-α-D-Man*p*-(1→, →2,6)-α-D-Man*p*-(1→3,6)-α-D-Man*p*-(1→, and →6)-β-D-Glc*p*-(1→2,6)-α-D-Man*p*-(1→, respectively. According to the NMR spectrum analysis, a possible structural unit of LPB8-1 was predicted and is depicted in Figure 3D.

### 3.5. Crystalline Features and Thermal Stability

XRD, a technique providing crystallinity information, has been widely used to examine the amorphous and crystalline structure of polysaccharides. As depicted in Figure 4A, LPB8-0 and LPB8-1 exhibited major crystalline reflections at 18.69° and 19.17°, respectively, confirming that they were semicrystalline polymers with low crystallinity. This was consistent with previous study results showing similar crystalline structures for the ESPs from *Bacillus licheniformis* PASS26 [23] and *B. cereus* KMS3-1 [24]. This special semicrystalline structure could be attributed to the order degree within the polysaccharides, which might directly affect the physical and functional properties of the polysaccharides, such as viscosity, solubility, water holding capacity, tensile strength, and swelling power.

Figure 4B depicts the thermal stability analysis results for LPB8-0 and LPB8-1, displaying two stages. In the first phase, an initial mass loss of approximately 5% was observed between 35 and 115 °C, which might have been ascribed to the evaporation and desorption of water. However, the mass remained unchanged from 115 to 240 °C, indicating that LPB8-0 and LPB8-1 were relatively stable below 240 °C. With the temperature increasing, the maximum mass loss was observed from 240 to 550 °C with a mass loss of 75%. This might have been due to the depolymerization of LPB8-0 and LPB8-1. Sharp peaks were observed on the DTG curves at 308.3 and 311.7 °C highlighting the high degradation temperature (Td) of LPB8-0 and LPB8-1, respectively, and especially of LPB8-1. The masses gradually decreased to 24.94% and 14.98% in LPB8-0 and LPB8-1, respectively, with an increase in temperature to 800 °C. The Td values of LPB8-0 and LPB8-1 were higher than those of the EPSs from *Leuconostoc pseudomesenteroides* DRP-5 (298.81 °C) [25] and *Lactobacillus sakei* L3 (272 °C) [26]. These results led to the conclusion that LPB8-0 and LPB8-1 had significant thermal stability, which is vital for food industries requiring a high level of thermal processing.

### 3.6. Particle Size and Zeta Potential Examination

Figure 4C,D presents the particle sizes (nm) of LPB8-0 and LPB8-1 (0.5% *w*/*v*) in aqueous solution. LPB8-0 and LPB8-1 possessed uniform particles, and the size of LPB8-0 particles (Figure 3C) ranged from 150 to 550 nm, with an average particle size of 286.3 ± 4.1 nm, while the size of LPB8-1 particles (Figure 3D) ranged from 100 to 450 nm, with an average particle size of 254.5 ± 2.3 nm. These particle sizes were smaller than those of the EPSs from *Lactobacillus plantarum* C70 (525.5 nm) [19] and *Pediococcus pentosaceus* M41 (446.8 nm) [27] but larger than that of the EPS produced by *Lactococcus garvieae* C47 (166.6 nm) [28]. The differences in particle size might be ascribed to the MW, linkages, and chemical compositions of the EPSs in question [19].

The zeta potentials (mV) of LPB8-0 and LPB8-1 solutions were detected, and the charges of LPB8-0 and LPB8-1 were −4.31 ± 0.75 mV and −18.3 ± 1.46 mV, respectively (Figure 4E,F). Compared with neutral EPSs, EPSs with negative charges exhibited stronger bioactivity and could alter the process of gel formation, thereby helping to improve the elasticity of the protein network [27]. Additionally, the physicochemical environment (pH, temperature), the range of MW, and charge density might affect the zeta potential [29].

### 3.7. Morphology Characteristics

As shown in Figure 5A,B, LPB8-0 and LPB8-1 appeared as a soft cotton-like structure. SEM revealed different surface morphologies of LPB8-0 and LPB8-1 (Figure 5C–F) at 1000× and 5000× magnifications. LPB8-0 exhibited an uneven surface primarily composed of an irregular, thin-sliced filiform structure, while the surface morphology of LPB8-1 was a branched, irregular, rod-like spherical body structure. The EPS produced by *Lactobacillus plantarum* HY [16] had a three-dimensional network structure combining sheets and tubes. The EPS obtained from *B. licheniformis* AG-06 [30] exhibited uneven smooth surfaces with network-like structures composed of irregular chains. The different shapes and structures might be attributable to the purification, preparation, and monosaccharide composition of the EPSs [31].

AFM images of LPB8-0 and LPB8-1 are depicted in Figure 5G–J. LPB8-0 was irregularly agglomerated in the dilute solution, suggesting molecular aggregation, while LPB8-1 was uneven in size and exhibited an irregularly spherical molecular conformation, which might have been ascribed to its branched structure and molecular aggregation through hydrogen bonds. This phenomenon was consistent with the SEM results. A similar spherical molecular conformation was found in the EPSs produced by other probiotic bacteria, such as *Lactobacillus plantarum* YW11 [32] and *B. megaterium* PFY-147 [33].

### 3.8. Emulsifying Activity and Emulsion Stability of the EPSs

#### 3.8.1. Emulsifying Activities with Several Edible Oils

The emulsifying activity and stability of the prepared emulsions were evaluated against several edible oils. As depicted in Figure 6A,B, after 1 h, LPB8-0 and LPB8-1 showed good EAs (100% ± 0.0%) against palm oil and olive oil, respectively. The LPB8-0 and LPB8-1 produced by *L. pentosus* B8 were found to possess good emulsion-stabilizing ability against several edible oils, as shown by the EA24 value, retaining at least 50% of the initial volume. An effective emulsifier maintains its capacity to the total emulsion volume 24 h (≥50%) after its formation [34]. Notably, the EA values (70.9% ± 1.2% and 79.9% ± 0.8%) of LPB8-0 and LPB8-1 against palm oil and olive oil remained high after 168 h. Compared with other EPS biopolymers, LPB8-0 and LPB8-1 from *L. pentosus* B8 had higher EA values. In previous reports, the EPS produced by *Virgibacillus salarius* BM02 had an EA24 value of 47.06% against olive oil [35]. By contrast, the EPS of *Bacillus coagulans* RK-02 had the highest emulsifying efficiency against sunflower seed oil (70%) than against soybean oil (62%) and rice oil (41%), at a similar concentration [36]. This phenomenon might be attributable to the specificity of the EPSs and certain hydrophobic compounds [35].

#### 3.8.2. Effects of EPS Concentration, pH, and Temperature on Emulsifying Activity

As an important criterion for many fields, an emulsifier should retain its stability when affected by various complicated conditions (concentration, pH, and temperature). As depicted in Figure 6C,D, the EA values significantly increased (*p* < 0.05) with increasing LPB8-0 and LPB8-1 concentrations (0.1–1 mg/mL). However, the emulsion showed good stability (*p* > 0.05) when it was exploited with LPB8-0 and LPB8-1 concentrations ranging from 1 to 2 mg/mL. Hence, 1 mg/mL was considered as the optimum concentration for LPB8-0 and LPB8-1 when used as an emulsifier. A similar concentration was observed for the EPS from *Bacillus amyloliquefaciens* LPL061 [37]. The emulsions of LPB8-0 and LPB8-1 against palm oil and olive oil exhibited excellent stability under different pH environments (Figure 6E). Nevertheless, a contradictory result, that is, a decrease in the EA24 value, was discovered under alkaline environments of pH 10 and 12. This result indicates that LPB8-0 and LPB8-1 could be potential emulsifiers under different pH environments. In a previous study, emulsions prepared using EPS22 and olive oil exhibited lower emulsion activity in the storage pH range of 10 to 12 (alkaline conditions) [34]. As depicted in Figure 6F, two emulsions retained good emulsifying activity in the temperature range of 25–100 °C. Another study reported that the activity of emulsions prepared with the EPS from *Virgibacillus salarius* BM02 and sunflower oil decreased to 47.06% at 100 °C [35]. These differences might be attributable to the excellent thermal stability of LPB8-0 and LPB8-1 at 308.3 and 311.7 °C, respectively, at pH 10 and 12.

### 3.9. Antioxidant Activities of LPB8-0 and LPB8-1

#### 3.9.1. DPPH Radical Scavenging Activities of LPB8-0 and LPB8-1

The scavenging abilities of LPB8-0 and LPB8-1 on DPPH free radicals are shown in Figure 7A. These two fractions exhibited satisfactory DPPH scavenging ability, and a positive correlation was observed between DPPH scavenging ability and the concentrations of LPB8-0 and LPB8-1. LPB8-1 exhibited higher scavenging ability than LPB8-0 at the concentrations of 0–10 mg/mL. When the concentrations reached 10 mg/mL, the DPPH radical scavenging rates of LPB8-0 and LPB8-1 were 50.62% ± 1.5% and 62.82% ± 2.0%, respectively. However, the scavenging abilities of these two fractions were significantly weaker than that of the control vitamin C (VC) (*p* < 0.05). Compared with LPB8-1, an EPS isolated from *Lactobacillus plantarum* CNPC003 exhibited a lower DPPH scavenging activity of 51.52 ± 1.10% at the concentration of 8 mg/mL [7].

#### 3.9.2. ABTS Radical Scavenging Activities of LPB8-0 and LPB8-1

Figure 7B depicts the ABTS free radical scavenging activities of LPB8-0 and LPB8-1. The ABTS free radical scavenging activities of these two EPS fractions were considerably lower than that of the VC, exhibiting a concentration-dependent activity. Compared with LPB8-1, LPB8-0 exhibited stronger scavenging activities against ABTS in the range of 0–10.0 mg/mL. This trend was similar to that of the DPPH free radical scavenging ability. At a concentration of 10 mg/mL, the scavenging activities of LPB8-0 and LPB8-1 against ABTS increased to 47.17% ± 1.7% and 58.36% ± 2.4%, respectively. In a previous study, the EPS from *Lactobacillus plantarum* JLAU103 showed an ABTS radical scavenging rate of 65.5% at a concentration of 10 mg/mL [38].

#### 3.9.3. Hydroxyl Free Radical Scavenging Abilities of LPB8-0 and LPB8-1

As depicted in Figure 7C, LPB8-0 and LPB8-1 exhibited a significant, concentration-dependent hydroxyl free radical scavenging activity in the concentration range of 0–10 mg/mL. Their scavenging activities were in the following order: VC > LPB8-1 > LPB8-0. The hydroxyl free radical scavenging capacities of 10 mg/mL LPB8-0 and LPB8-1 were 47.91% ± 1.7% and 72.52% ± 2.7%, respectively. Their hydroxyl free radical scavenging activities were stronger than those of the EPSs from *Lactobacillus rhamnosus* ZFM231 (49.7%) [6] and *Leuconostoc mesenteroides* DRP105 (30.48% ± 0.78%) [39]. Overall, these results indicated that these two polymers (LPB8-0 and LPB8-1) could be potential alternatives to synthetic antioxidants, especially LPB8-1.

#### 3.9.4. Correlation between Structure and Antioxidant Activity

Research fronts have proven that polysaccharides possess excellent antioxidant efficacy, but their antioxidant mechanism is still obscure. Zhang et al. [40] discovered that the antioxidant activity pathway of polysaccharides might be ascribed to the presence of OH groups that could improve hydrogen to stabilize the free radicals or directly react with the free radicals to terminate the radical chain reaction. The antioxidant abilities of polysaccharides largely depend on their purity, monosaccharide compositions, MWs, amounts and positions of functional groups, glycosidic linkages, and chain conformations. The purities of LPB8-0 and LPB8-1 were 96.2% ± 1.0% and 99.1% ± 0.5%, respectively. However, LPB8-1 exhibited better antioxidant activity than LPB8-0. Jiang et al. [41] reported that the polysaccharide (MBP-2) extracted from mung bean skin with higher carbohydrate content (89.26%) exhibited stronger antioxidant activity than that with lower carbohydrate content (MBP-1; 86.69%). These results were in agreement with those reported by Xie et al. [42]. Moreover, the antioxidant properties of the polysaccharides studied herein were closely related to their MWs. In this study, LPB8-1, with a high MW, exhibited better antioxidant properties, indicating that EPSs with higher MWs possess stronger antioxidant activities, which would be consistent with the results of a previous study [43]. However, polysaccharides produced by *Sagittaria sagittifolia* L. with lower MWs (16.62 kDa) had better antioxidant abilities, which might have been due to the fact that low-MW polysaccharides could accept more free radicals [44]. The inconsistency in the aforementioned results indicates that MW does not affect the antioxidant properties of polysaccharides. An et al. [45] reported the significant effects of monosaccharide composition on antioxidant activity. In previous studies, some monosaccharides (mannose, galactose, xylose, rhamnose, arabinose, etc.), the major constituents of polysaccharides, exhibited satisfying antioxidant activity [46,47]. Thus, the high proportions of mannose and galactose in LPB8-1 might have contributed to its high antioxidant efficacy. Furthermore, Feng et al. [48] found that the antioxidant activity of polysaccharides might be governed by their functional groups (hydroxyl and carboxyl) and chain conformation. In this study, we inferred that the antioxidant activity of LPB8-0 and LPB8-1 might have depended not on one factor but rather on a combination of multiple factors (high purity, high MW, and high mannose and galactose contents). However, further exploration is highly recommended to explore the detailed antioxidant mechanism of EPSs produced by LAB.

## 4. Conclusions

In this study, two novel EPSs (LPB8-0 and LPB8-1) were isolated and purified from *L. pentosus* B8. The MWs, monosaccharide compositions, and surface morphologies of LPB8-0 and LPB8-1 were slightly different, with typical absorption peaks of polysaccharides. The backbone of LPB8-1 was composed of →2)-α-D-Man*p*-(1→ and →6)-α-D-Man*p*-(1→. The results demonstrated that LPB8-0 and LPB8-1 were semicrystalline polymers with outstanding thermal stability. LPB8-0 and LPB8-1 in aqueous solution had average particle diameters of 286.3 ± 4.1 nm and 254.5 ± 2.3 nm and negative charges of −4.31 ± 0.75 mV and −18.3 ± 1.46 mV, respectively. Both LPB8-0 and LPB8-1 showed excellent emulsifying activities against several edible oils, especially palm oil and olive oil. Additionally, DPPH, ABTS, and hydroxyl free radical assays showed that LPB8-0 and LPB8-1 exhibited potential antioxidant activities in vitro in a concentration-dependent manner. LPB8-1, with high purity and a high MW, had better antioxidant ability. Overall, the study results provide a theoretical basis for the potential applications of LBP8-1 as an emulsifier and antioxidant in foods and pharmaceuticals.

## Figures and Tables

**Figure 1 foods-11-02327-f001:**
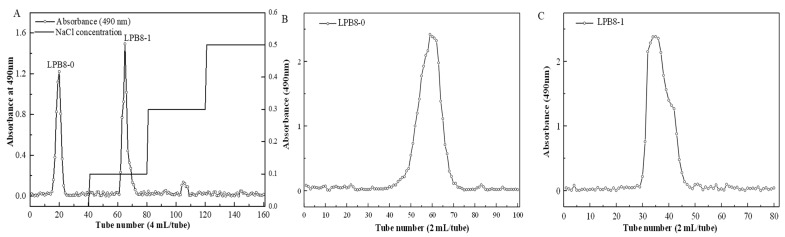
(**A**) DEAE-52 anion-exchange chromatogram; (**B**) Sephacryl S-300 HR chromatographic profile of LPB8-0; (**C**) Sephacryl S-400 HR chromatographic profile of LPB8-1.

**Figure 2 foods-11-02327-f002:**
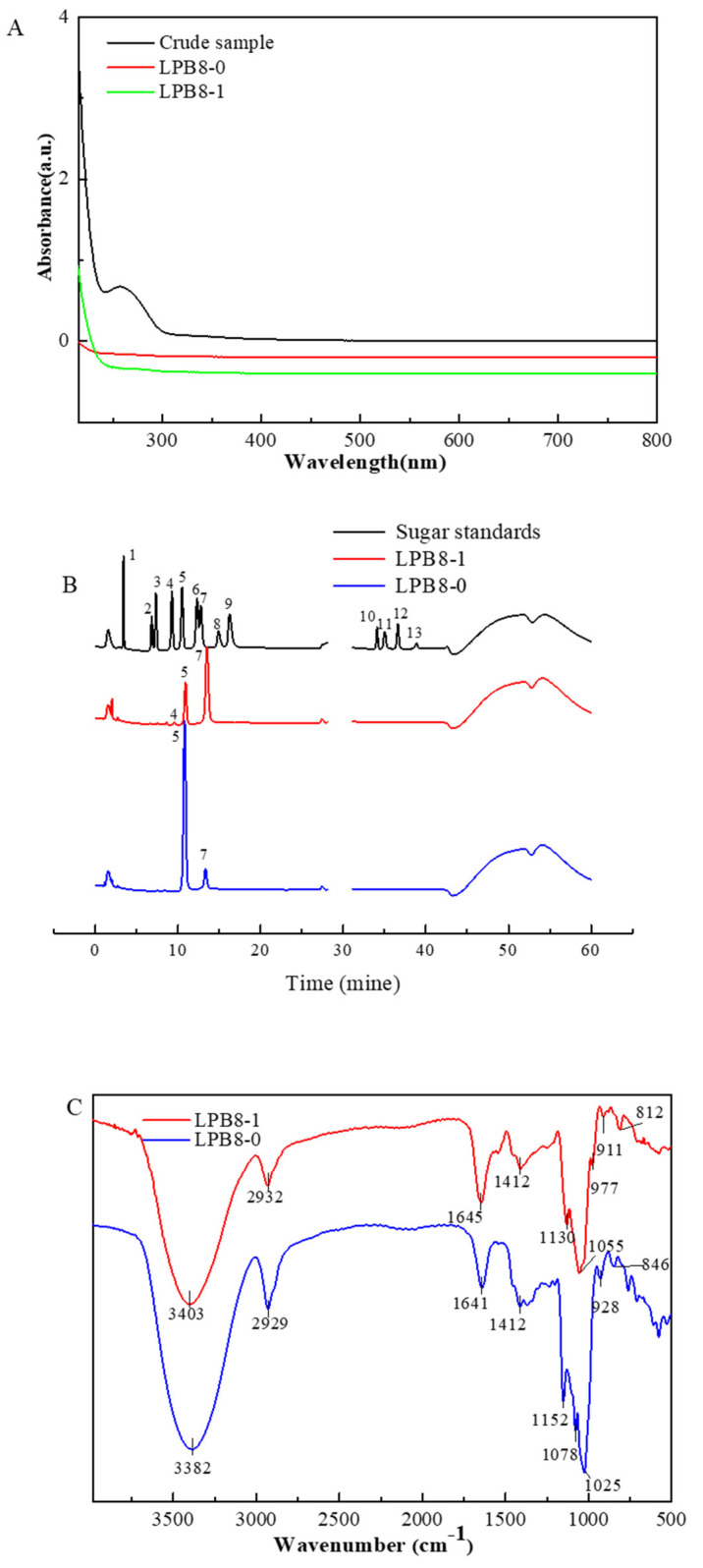
(**A**) UV−vis absorption spectra of crude EPS (black curve), LPB8-0 (red curve), and LPB8-1 (green curve); (**B**) HPAEC-PAD profiles of monosaccharide standards (black curve, peak identities: 1, fucose; 2, rhamnose; 3, arabinose; 4, galactose; 5, glucose; 6, xylose; 7, mannose; 8, fructose; 9, ribose; 10, galacturonic acid; 11, guluronic acid; 12, glucuronic acid; 13, mannuronic acid), LPB8-0 (blue curve) and LPB8-1 (red curve); (**C**) FTIR spectra of LPB8-0 (blue curve) and LPB8-1 (red curve).

**Figure 3 foods-11-02327-f003:**
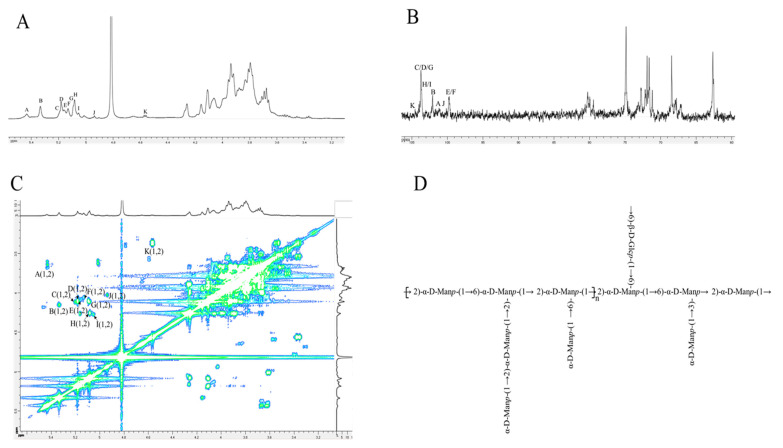
NMR spectra of LPB8-1 recorded in D_2_O at 298 K: (**A**) 1D-1H spectrum; (**B**) 1D-13C spectrum; (**C**) COSY spectrum; (**D**) one putative structure of LPB8-1.

**Figure 4 foods-11-02327-f004:**
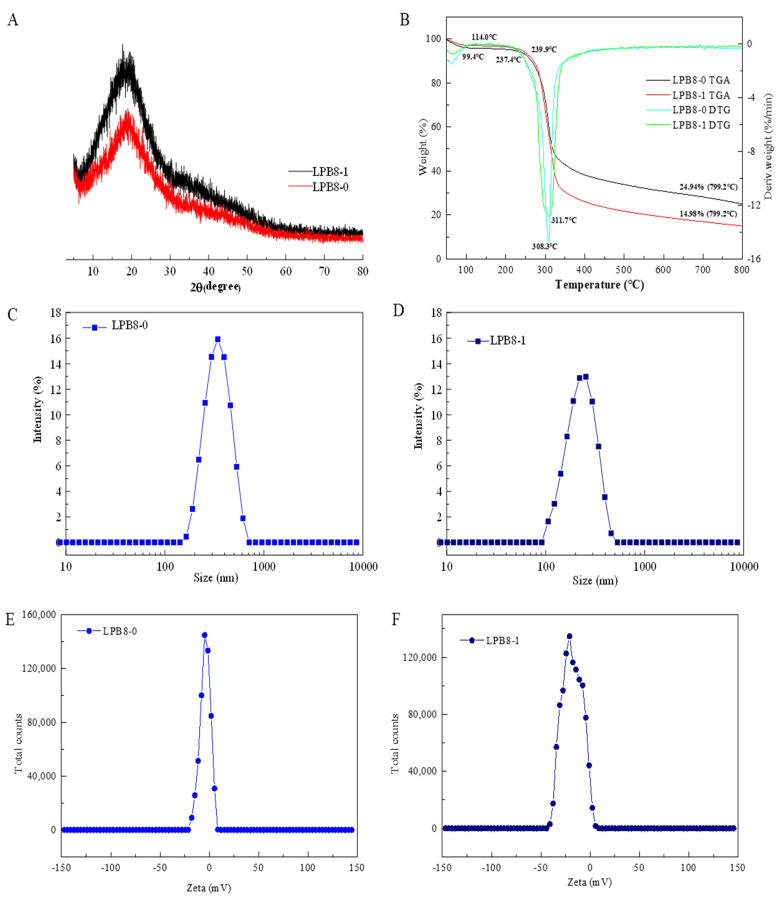
The XRD (**A**) and TGA (**B**) spectra of LPB8-0 and LPB8-1; the zeta potential (**C**,**D**) and particle size (**E**,**F**) at various concentrations of LPB8-0 and LPB8-1.

**Figure 5 foods-11-02327-f005:**
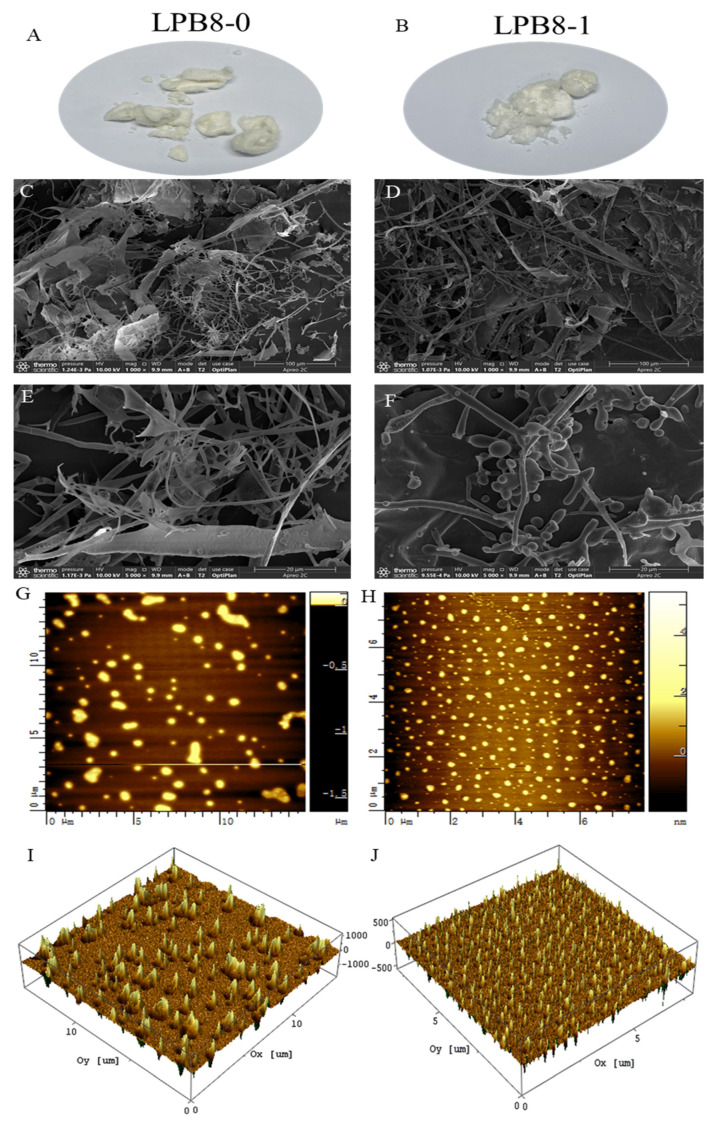
The appearances of LPB8-0 (**A**) and LPB8-1 (**B**); SEM images of LPB8-0 and LPB8-1 under 1000× (**C**,**D**) and 5000× (**E**,**F**) magnification; AFM showing the topographic features of LPB8-0 and LPB8-1, planar view (**G**,**H**) and cubic view (**I**,**J**).

**Figure 6 foods-11-02327-f006:**
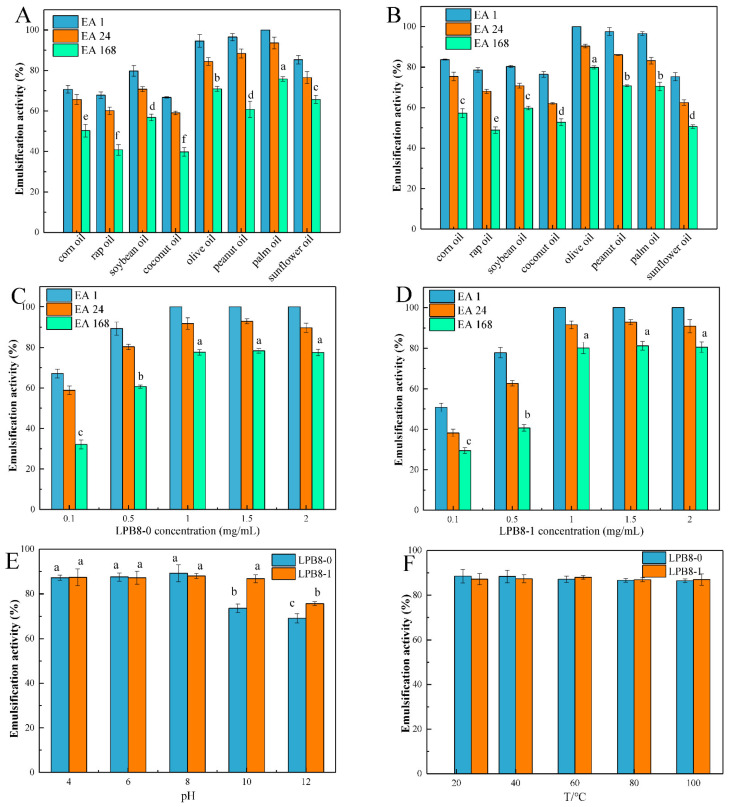
Emulsification activity of emulsions prepared with different oils (**A**,**B**) and different LPB8-0 and LPB8-1 concentrations (**C**,**D**); influence of pH (**E**) and temperature (**F**) on the emulsification activity of the emulsions. Statistical differences are indicated with different lowercase letters (*p* < 0.05).

**Figure 7 foods-11-02327-f007:**
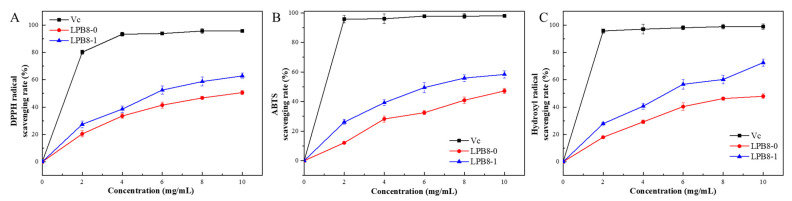
The antioxidant activities of LPB8-0 and LPB8-1 using VC as a control: scavenging activity towards DPPH (**A**); ABTS radical (**B**); hydroxyl radical (**C**).

**Table 1 foods-11-02327-t001:** Chemical compositions and basic properties of LPB8-0 and LPB8-1.

Factions	Carbohydrate Content (%)	Protein Content (%)	Sulfates (%)	Mw (Da)
LPB8-0	96.2 ± 1.0%	Nd *	Nd *	1.12 × 10^4^
LPB8-1	99.1 ± 0.5%	Nd *	Nd *	1.78 × 10^5^

* Not detected.

**Table 2 foods-11-02327-t002:** Glycosidic linkage composition of methylated LPB8-1 by GC–MS analysis.

Time (min)	Methylated Sugars	Deduced Linkages	Molar Ratios
8.9	2,3,4,6-Me_4_-Man*p*	T-Man*p*-(1→	36.00
12.3	2,4,6-Me_3_-Man*p*	→3)-Man*p*-(1→	11.94
12.4	3,4,6-Me_3_-Man*p*	→2)-Man*p*-(1→	17.31
13.6	2,3,4-Me_3_-Man*p*	→6)-Man*p*-(1→	2.58
13.7	2,3,4-Me_3_-Glc*p*	→6)-Glc*p*-(1→	2.93
14.1	2,3,6-Me_3_-Glc*p*	→4)-Glc*p*-(1→	11.11
17.9	2,4-Me_2_-Man*p*	→3,6)-Man*p*-(1→	0.97
18.2	3,4-Me_2_-Man*p*	→2,6)-Man*p*-(1→	10.10
18.2	3,4-Me_2_-Glc*p*	→2,6)-Glc*p*-(1→	6.74
21.4	4-Me-Man*p*	→2,3,6)-Man*p*-(1→	0.32

**Table 3 foods-11-02327-t003:** ^1^H and ^13^C NMR chemical shift data for LPB8-1.

Sugar Residue	Chemical Shifts δ (ppm)						
	H-1/C-1	H-2/C-2	H-3/C-3	H-4/C-4	H-5/C-5	H-6/C-6	H-6′
A: →4)-α-D-Glc*p*-(1→	5.43/101.28	3.66/72.05	4.01/72.64	3.67/73.08	3.83/74.25	3.93/61.84	3.69
B: →2)-α-D-Man*p*-(1→	5.33/102.29	4.16/79.80	3.96/71.68	3.79/68.01	3.92/74.49	3.80/62.38	3.66
C: →3)-α-D-Man*p*-(1→	5.19/103.62	4.11/71.36	3.92/74.74	3.68/68.24	3.85/74.64	3.94/62.46	3.70
D: α-D-Man*p*-(1→	5.18/103.54	4.10/71.29	3.91/72.07	3.67/68.24	3.82/74.72	3.93/62.23	3.69
E: →2,6)-α-D-Man*p*-(1→	5.16/99.72	4.07/80.27	3.80/71.91	3.70/68.55	3.83/74.57	3.92/68.08	3.78
F: →2,6)-α-D-Man*p*-(1→	5.13/99.64	4.06/80.19	3.80/71.76	3.71/68.63	3.82/74.62	3.93/68.32	3.78
G: α-D-Man*p*-(1→	5.09/103.39	4.10/71.44	3.91/71.68	3.68/68.32	3.81/74.57	3.92/62.46	3.69
H: →3,6)-α-D-Man*p*-(1→	5.08/103.62	4.28/71.05	3.89/75.05	3.78/67.85	3.95/74.72	3.84/68.01	3.79
I: →6)-α-D-Manp-(1→	5.05/103.54	4.28/71.21	3.97/71.76	3.79/67.85	3.95/74.41	3.82/68.16	3.77
J: →2,6)-β-D-Glc*p*-(1→	4.93/101.04	4.03/79.72	3.65/71.32	3.91/70.82	3.80/74.18	3.91/68.24	3.71
K: →6)-β-D-Glc*p*-(1→	4.56/104.56	3.37/74.64	3.54/77.14	3.67/69.26	3.78/74.80	3.97/67.77	3.62

## Data Availability

Data is contained within the article or Appendix A.

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
