# Peer review of "Extraction, Structural Analysis, and Biofunctional Properties of Exopolysaccharide from Lactiplantibacillus pentosus B8 Isolated from Sichuan Pickle"

_foods, 2022, doi:10.3390/foods11152327_

Round 1

Reviewer 1 Report

This study has investigated the extraction, structural analysis, and biofunctional properties of two exopolysaccharides obtained from from Lactiplantibacillus pentosus B8 isolated from Sichuan pickle. The manuscript is well structured and the results are very interesting.

Comments:  3.1. Extraction, purification, and chemical composition of EPS: the explanations about extraction and purification method should be transferred to materials and methods Line 170: Add the mode (contact, tapping ,...), magnification and the details of the AFM imaging. Line 193: production yield Line 323: why the EPSs had negative charge?  

Author Response

Reviewer 1 This study has investigated the extraction, structural analysis, and biofunctional properties of two exopolysaccharides obtained from from Lactiplantibacillus pentosus B8 isolated from Sichuan pickle. The manuscript is well structured and the results are very interesting.

Response: Thanks a lot for appreciating our work.

Comments:  3.1. Extraction, purification, and chemical composition of EPS: the explanations about extraction and purification method should be transferred to materials and methods.

Response: Thank you so much for your valuable suggestion. In the materials and methods part, we have described the process of the extraction and purification in detail (Lines 91-112). In section 3.1, we mainly described the results of extraction, purification, and chemical composition, but not to repeat the methods. For example, two fractions were obtained (Fig 1A), the homogeneous of LPB8-0 and LPB8-1 (Fig 1B, C) and the chemical composition of LPB8-0 and LPB8-1 (Table 1) were showed, no nucleic acids or proteins in LPB8-0 and LPB8-1 were indicated (Fig 2A). The aim of using those sentences describing what has been done to the samples was to further introduce the subsequent results, and meanwhile keep the logic of the whole paragraph. We believe that this kind of description is widely used in a large number of published articles [Hu et al. (2020) Carbohydrate Polymers, 240, 116238; Gao et al. (2020) Carbohydrate Polymers, 249, 116874; Jiang et al. (2020) Food Hydrocolloids 100, 105412; Liu et al. (2020) LWT - Food Science and Technology, 17,108645; Liao et al. (2022) International Journal of Biological Macromolecules 195, 466-474; Shen et al. (2022) International Journal of Biological Macromolecules 199, 24-35].

Line 170: Add the mode (contact, tapping ,...), magnification and the details of the AFM imaging.

Response: Thanks for pointing out this. We have added related contents according to your suggestion (Lines 167-169).

Line 193: production yield

Response: We have made correction according to your suggestion (Line 221).

Line 323: why the EPSs had negative charge? 

Response: The charge of a substance could reflect the stability of solution or colloid, which directly decides the substance’s potential application. The small amount of negative charges on the polysaccharides could suggest that samples were neutral hydro soluble compounds [Li et al. (2016) International Journal of Biological Macromolecules, 83, 270-276]. Based on the results of the monosaccharide composition, no uronic acid was found in LPB8-0 and LPB8-1, which showed LPB8-0 and LPB8-1 were neutral polysaccharides. In addition, the negative charge also indicated that the polysaccharides possessed the donated electron capacities. Similar result has also been obtained in previous studies [Chen et al. (2020) International Journal of Biological Macromolecules, 155, 674-684; Gan et al. (2021) Food Chemistry, 365, 130496, Wang et al. (2019) Carbohydrate Polymers, 211, 227-236]

Reviewer 2 Report

Lactiplantibacillus pentosus B8 (L. pentosus B8) was isolated from Sichuan pickle as a new strain producing exopolysaccharide (EPS), and two EPSs (designated as LPB8-0 and LPB8-1) were purified from L. pentosus B8. Then, the structural and bio-functional properties of these two EPSs were investigated, and the results of characterization were presented.

 The experiments were designed and carried out without any serious fault, and the analysis and interpretation of obtained results were considered to be reasonable and appropriate. However, the domain of this research were considered to be relatively unique and specific, and therefore the description and explanation of the experimental procedures and results would hopefully be more clearer, and easy to imagine and understand the actual manipulation in detail.

 The manuscript, especially the wording, would be polished up a little more to avoid unnatural or odd words and expression. Also, the structures and the word orders of some sentences should be revised.

Author Response

Reviewer 2  Lactiplantibacillus pentosus B8 (L. pentosus B8) was isolated from Sichuan pickle as a new strain producing exopolysaccharide (EPS), and two EPSs (designated as LPB8-0 and LPB8-1) were purified from L. pentosus B8. Then, the structural and bio-functional properties of these two EPSs were investigated, and the results of characterization were presented.

 The experiments were designed and carried out without any serious fault, and the analysis and interpretation of obtained results were considered to be reasonable and appropriate.

Response: Thanks a lot for appreciating our work.

However, the domain of this research were considered to be relatively unique and specific, and therefore the description and explanation of the experimental procedures and results would hopefully be more clearer, and easy to imagine and understand the actual manipulation in detail.

Response: Thank you so much for your valuable suggestion. We have made correction according to your suggestion (Lines 167-169; Lines 180-212).

 The manuscript, especially the wording, would be polished up a little more to avoid unnatural or odd words and expression. Also, the structures and the word orders of some sentences should be revised.

Response: Thank you so much for your constructive comments on our manuscript. For proper English language, the revised manuscript has been re-edited throughout.

Reviewer 3 Report

The synthesis of a new substances with antioxidant properties and emulsifiers is very important in the context of their possible use in the food or pharmaceutical industry. The research objectives are stated clearly but from my point of view, this paper requires major revisions. Manuscript, in accordance with the journal's guidelines, should be sent in Microsoft Word template or LaTeX template. The graphics of some figures are insufficient, and descriptions of research methods also need to be supplemented. Further detailed comments for consideration are provided below.

Comment 1#

·         The manuscript is not formatted according to the journal guidelines  …. use the Microsoft Word template or LaTeX template to prepare your manuscript !!!!

·         Line 182 page 9 DPPH radical scavenging ability, ABTS+ free radical scavenging ability, Hydroxyl free radical scavenging ability …… the methodology for antioxidant determination requires a more detailed description. The lack of calculation methods and the implementation procedure

·         Line 307 page 14 Leuconostoc…. incorrect font size

·         Line 318 page 15 Lactobacillus... Pediococcus ….…. incorrect font size

·         Line 319 page 15 Lactococcus….. incorrect font size

·         Fig 2 A is Curde sample ……it should be…. Crude sample

·         Fig 6 the graphics of the charts are not readable, especially C,D,E,F

·         Table 1 is Carbuhydrate content….. it should be…….. Carbohydrate content

Author Response

Reviewer 3 The synthesis of a new substances with antioxidant properties and emulsifiers is very important in the context of their possible use in the food or pharmaceutical industry. The research objectives are stated clearly but from my point of view, this paper requires major revisions. Manuscript, in accordance with the journal's guidelines, should be sent in Microsoft Word template or LaTeX templateThe graphics of some figures are insufficient, and descriptions of research methods also need to be supplemented. Further detailed comments for consideration are provided below.

Comment 1#

  • The manuscript is not formatted according to the journal guidelines  ….use the Microsoft Word template or LaTeX template to prepare your manuscript !!!!
  • Line 182 page 9 DPPH radical scavenging ability, ABTS+ free radical scavenging ability, Hydroxyl free radical scavenging ability …… the methodology for antioxidant determination requires a more detailed description. The lack of calculation methods and the implementation procedure.

Response: Thank you so much for your valuable suggestion. More detailed descriptions have been (Lines 180-212).

  • Line 307 page 14 Leuconostoc…. incorrect font size
  • Line 318page 15 Lactobacillus... Pediococcus ….…. incorrect font size
  • Line 319page 15 Lactococcus….. incorrect font size

Response: Thank you very much for your careful investigation throughout the whole text. The mistake you mentioned has been corrected (Line 334; Line 344; Line 346).

  • Fig 2 A is Curde sample ……it should be…. Crude sample

Response: We have made correction according to your suggestion (Fig 2A).

  • Fig 6 the graphics of the charts are not readable, especially C,D,E,F

Response: We have made correction according to your suggestion. Improved figures were attached in the file “figures” (Fig 6; Fig 7).

  • Table 1 is Carbuhydrate content….. it should be…….. Carbohydrate content

Response: Thanks for pointing out this. We have made correction according to your suggestion (Table 1).

Round 2

Reviewer 3 Report

Manuscript, in accordance with the journal's guidelines, should be sent in Microsoft Word template or LaTeX template, see point 2 of Submission Checklist (Instructions for Authors). If the Editors do not require this condition to be met, I have no objection to the manuscript.